# Deterioration of contrast sensitivity in eyes with epiphora due to lacrimal passage obstruction

Kuniharu Tasaki[☯], Sujin Hoshi[ID]*[☯], Takahiro Hiraoka, Tetsuro Oshika

Department of Ophthalmology, Faculty of Medicine, University of Tsukuba, Ibaraki, Japan

☯ These authors contributed equally to this work.
* hoshisujin@md.tsukuba.ac.jp

## Abstract

### Purpose

Epiphora causes deterioration in contrast sensitivity in some eye diseases. This study was conducted to investigate contrast sensitivity in eyes with epiphora caused by lacrimal passage obstruction.

### Methods

This single-center, prospective case series enrolled 57 patients with unilateral lacrimal passage obstruction. The best-corrected visual acuity (BCVA), contrast sensitivity function, and lower tear meniscus of the affected and contralateral unaffected eyes were compared. The area under the log contrast sensitivity function (AULCSF) was calculated.

### Results

The BCVA did not significantly differ between the affected and contralateral eyes, while the AULCSF was significantly lower in the affected eyes than that in the contralateral eyes (median 1.35, interquartile range 1.22–1.44 vs. median 1.36, interquartile range 1.28–1.46, $P = 0.032$). Lower tear meniscus parameters were significantly higher in the affected eyes than those in the contralateral eyes ($P < 0.005$).

### Conclusions

The contrast sensitivity function is significantly diminished in eyes with epiphora caused by lacrimal passage obstruction.

## Introduction

Patients with epiphora caused by lacrimal passage obstruction often complain of vision-related symptoms. Although the decrease in visual function is rarely detected by conventional visual

**Data Availability Statement:** All relevant data are within the manuscript.

**Funding:** The author(s) received no specific funding for this work.

**Competing interests:** The authors have declared that no competing interests exist.

acuity measurements, several studies have assessed and revealed the negative effects of epiphora on the patient's quality of vision (QoV) and optical quality. QoV is assessed by measuring functional visual acuity (FVA), which demonstrates the change in visual acuity (VA) over time [1,2]. Patients with lacrimal passage obstruction experience significant deterioration in FVA, and an increase in higher-order aberrations until they undergo lacrimal passage intubation [3]. The vision-related quality of life is also significantly impaired before silicone tube intubation for lacrimal passage obstruction, composite score from the 25-item national eye institute visual function questionnaire (NEIVFQ-25) improved 76.3 ± 11.5 to 82.0 ± 11.3 (p = 0.001) after silicone tube intubation [4].

Contrast sensitivity, which is the ability to detect differences in luminance between adjacent areas, is a fundamental feature of vision. The measurement of contrast sensitivity provides useful information about QoV that may not be obtained by standard VA testing [5–9]. Contrast sensitivity is reduced in epiphora induced by conjunctivochalasis [10] or instillation of a gel-forming solution or particle suspension [11]. As a result, we hypothesize that contrast sensitivity will also be reduced in eyes with epiphora due to lacrimal passage obstruction. Thus, we aimed to assess contrast sensitivity as a measure of QoV, using different methods in adult eyes with lacrimal passage obstruction.

## Materials and methods

This study was a single institutional prospective case series, approved by the institutional review board of the University of Tsukuba Hospital (H27-153), and adhered to the tenets of the Declaration of Helsinki. The nature and possible consequences of the study were explained in detail, following which all patients provided informed consent.

### Patient population

Patients with unilateral lacrimal passage obstruction who visited the University of Tsukuba Hospital between November 2015 to July 2019 and had a best-corrected distance VA of 20/20 or better in both eyes measured by Snellen testing were considered for enrollment. The inclusion criteria were symptoms of epiphora, and the presence of at least one of the following dacryoendoscopic findings: nasolacrimal duct obstruction, canalicular obstruction, or punctual obstruction. The exclusion criteria were congenital lacrimal duct obstruction, acute dacryocystitis, and a history of ocular surface surgery. Patients with cortical cataract formation in the central lens, unilateral intraocular lens implantation, anisocoria, other ocular diseases, or a history of treatment that might affect contrast sensitivity were also excluded. Patients receiving ophthalmic solution stopped using ophthalmic solution 4 weeks before examinations. A total of 57 patients (men: 17, women: 40; mean age: 60.0 ± 11.4 years) with unilateral lacrimal passage obstruction participated in our study. Table 1 presents the classification of the

**Table 1. Type of obstruction based on dacryoendoscopy.**

| Underlying Disease | Cases (N) |
|---|---|
| Nasolacrimal duct obstruction | 34 |
| Common canalicular obstruction combined with nasolacrimal duct obstruction | 14 |
| Common canalicular obstruction | 5 |
| Upper and lower punctal obstruction | 1 |
| Upper and lower canalicular obstruction | 1 |
| Upper and lower punctal obstruction combined with nasolacrimal duct obstruction | 1 |
| Upper and lower canalicular obstruction combined with nasolacrimal duct obstruction | 1 |

types of obstruction diagnosed in all the participants. The patients presented with epiphora for a median duration of 9 (range 2–82, interquartile range 3–20.5) months.

## Assessment of tear meniscus

Cross-sectional images of the lower tear meniscus were captured vertically across the central cornea using swept-source anterior segment optical coherence tomography (OCT) (SS-1000, CASIA; Tomey Corp, Nagoya, Japan). The OCT images were processed using the in-built software. The principles, technique, and reproducibility of this device for evaluating tear meniscus have been described previously [12,13]. Lower tear meniscus height (TMH) and lower tear meniscus area (TMA) were calculated from the cross-sectional OCT images of the lower tear meniscus. The measurement was performed between 4 and 5 seconds after blinking, with spontaneous eye-opening.

## Assessment of contrast sensitivity

We measured three indices of contrast sensitivity function: contrast sensitivity (using the CSV-1000E chart), low-contrast VA (using the CSV-1000LanC10% chart), and letter contrast sensitivity (using the CSV-1000LV chart), which were obtained from Vector Vision CO., Greenville, OH, U.S.A. The pupils of the eyes were undilated during these monocular tests, and these tests were performed under photopic conditions. The testing distance was 2.5 m, with the best spectacle correction. The fluorescent luminance source of the instrument, which was automatically calibrated to 85 cd/m$^2$, provided background illumination for the translucent chart.

The CSV-1000E chart has vertical sine-wave gratings at four spatial frequencies, i.e., 3, 6, 12, and 18 cycles/degree, and each spatial frequency has eight different levels of contrast. Each row consists of eight pairs of circular patches and includes sine waves of a single spatial frequency. One patch of each pair presents a grating, and the other patch is blank. The patients were asked to identify the patch with the grating, and the contrast level of the last correct response was defined as the contrast threshold in logarithmic values for each frequency [14]. The area under the log contrast sensitivity function (AULCSF) was calculated from these data, according to the method described by Applegate et al. [15]. The AULCSF was determined as the integration of the fitted third-order polynomials of the log contrast sensitivity units between the fixed limits of 0.48 (corresponding to three cycles/degree) and 1.26 (18 cycles/degree) on the log spatial frequency scale. This presents contrast sensitivity data as one number and makes statistical analysis easier.

The CSV-1000LanC10% chart uses the Landolt ring as the optotype under 10% low-contrast. There are five letters per line, and each one-line step represents a change of 0.1 logMAR units. Low-contrast VA was scored by assigning a value of 0.02 logMAR units for each correctly identified letter.

The CSV-1000LV chart uses letter optotypes. Each letter is of the same size and is of low spatial frequency (2.4 cycles/degree). There are eight contrast levels (standard, 35.5%, 17.8%, 8.9%, 6.3%, 4.5%, 2.2%, and 1.1%), and each contrast level has three letters. The test results were recorded as the number of correctly identified letters and not as the contrast sensitivity or contrast threshold [16,17].

## Statistical analyses

Normally distributed data obtained from the affected eyes were compared with the contralateral eyes using the paired t-test (two-tailed test). Data that were not normally distributed were compared using the Wilcoxon signed-rank test. Analysis of the correlation between the

Table 2. Comparison between the measured parameters for the affected and contralateral eyes.

| Parameters | Affected Eyes | Contralateral Eyes | P Value |
|---|---|---|---|
| [a]BCVA (logMAR) | -0.1 ± 0.05 | -0.11 ± 0.06 | 0.341 |
| [b]AULCSF | 1.35 (1.22–1.44) | 1.36 (1.28–1.46) | 0.032 |
| [a]Low-contrast visual acuity (logMAR) | 0.18 ± 0.18 | 0.16 ± 0.04 | 0.654 |
| [a]Letter contrast sensitivity (number of letters) | 23 ± 1.1 | 22.8 ± 1.3 | 0.314 |
| [a]Tear meniscus height (mm) | 0.52 ± 0.21 | 0.33 ± 0.14 | < 0.005 |
| [a]Tear meniscus area (mm$^2$) | 0.1 ± 0.09 | 0.04 ± 0.03 | < 0.005 |

[a]Data are presented as mean ± SD. *P* value is evaluated using paired t-test.

[b]Data are expressed as median (interquartile range). *P* value is evaluated using Wilcoxon signed-rank test.

BCVA, best-corrected visual acuity; AULCSF, area under the log contrast sensitivity function.

difference in tear meniscus dimension (TMH and TMA) and the difference in QoV (BCVA and contrast sensitivity) of the affected and contralateral eyes were evaluated using Pearson correlation coefficient. *P* values < 0.05 were considered significant for all analyses. Statistical analysis was performed using Statcel (add-in software for Microsoft Excel), version 4 (Microsoft Corp., Redmond, WA).

## Results

### Comparison of parameters between affected and contralateral eyes

Table 2 shows the comparison of the BCVA, AULCSF, low-contrast VA, letter contrast sensitivity, TMH, and TMA for the affected and contralateral eyes. BCVA was comparable for both eyes, while AULCSF was significantly lower in the affected eye when compared to the contralateral eye (*P* = 0.032, Wilcoxon signed-rank test). Fig 1 shows the contrast sensitivity at four specific frequencies for the affected and contralateral eyes. There was a significant difference in

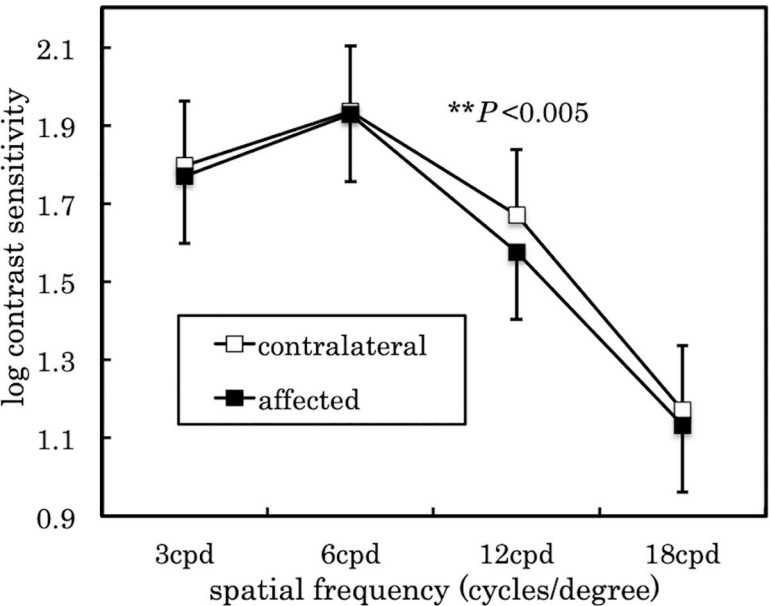

**Fig 1. Contrast sensitivity at four specific frequencies in the affected and contralateral eyes.** There was a significant difference in contrast sensitivity at 12 cycles/degree (*P < 0.005; paired t-test). Values are expressed as mean ± SD.

contrast sensitivity at 12 cycles/degree ($P < 0.005$, paired t-test), while there was no significant difference in contrast sensitivity at the other three spatial frequencies.

No significant differences were observed in the low-contrast VA ($P = 0.654$, paired t-test) and letter contrast sensitivity ($P = 0.314$, paired t-test) between the affected and contralateral eyes. The lower tear meniscus parameters determined by the TMH and TMA were significantly higher in the affected eye, when compared to the contralateral eye ($P < 0.005$, paired t-test).

## Correlations between the difference in tear meniscus and the difference in quality of vision of affected and contralateral eyes

The difference in AULCSF of the affected and contralateral eyes did not correlate with the difference in TMH ($r = -0.09$, $p = 0.491$) or TMA ($r = 0.04$, $p = 0.784$). The difference in log contrast sensitivity in 12cpd was neither correlated to the difference in TMH ($r = 0.01$, $p = 0.931$) and in TMA ($r = 0.09$, $p = 0.502$). As for other contrast sensitivity measurements, the difference in Low-contrast visual acuity was not correlated to the difference in TMH ($r = 0.09$, $p = 0.498$) and in TMA ($r = -0.02$, $p = 0.862$). Letter contrast sensitivity was negatively correlated to TMH ($r = -0.26$, $p = 0.044$) but was not correlated to TMA ($r = -0.20$, $p = 0.132$). The difference in BCVA was not correlated to the difference in TMH ($r = 0.12$, $p = 0.358$) and in TMA ($r = 0.07$, $p = 0.601$).

## Discussion

To the best of our knowledge, this is the first study to report a reduction in contrast sensitivity in eyes with epiphora caused by unilateral lacrimal passage obstruction. Although there were no significant differences in the conventional BCVA between eyes with epiphora and the contralateral eyes; the contrast sensitivity calculated as AULCSF was significantly decreased in the eyes with lacrimal passage obstruction and epiphora.

Contrast sensitivity, which is related to the deterioration in the quality of vision, is reduced in various ocular diseases, such as cataract [18,19], glaucoma [20], vitreous floaters [21], age-related macular degeneration [20,22], and neuromyelitis optica spectrum disorders [23], and also in refractive abnormalities such as LASIK intervention [24,25], myopia [26], and multifocal contact lenses [27,28]. How well a patient sees at the higher spatial frequency channels, which is well captured by conventional measures of visual acuity, does not necessarily predict vision at middle and lower frequencies [29]. For example, increased higher-order aberrations in eyes with LASIK intervention cause deterioration of contrast sensitivity while visual acuity is not affected [25]. Contrast sensitivity tests are more sensitive than standard VA testing and should be performed in patients with various tears/corneal abnormalities. Since a decline in VA is rarely detected in patients with epiphora caused by lacrimal passage obstruction, it is crucial to assess contrast sensitivity function to understand the link between the symptoms and their impact on the quality of life. Moreover, activities related to the quality of life, including driving [19,30], mobility, walking speed [31,32], reading speed [33,34], and computer task accuracy [35] are reportedly associated with contrast sensitivity. Patients with epiphora often have difficulties in driving, descending stairs, reading books, or text on computer monitors, which might be affected by deterioration of contrast sensitivity.

Deterioration of contrast sensitivity occurs in dry eye disease, a common disease characterized by abnormalities in the tear film on the ocular surface. Koh et al. reported that AULCSF decreased to $1.24 \pm 0.16$ in patients with dry eye, while that of normal eyes was $1.35 \pm 0.11$ (the participants' ages were approximately 50 years in both groups) [36]. In our study, the AULCSF was $1.35 \pm 0.15$ in the unaffected eyes, which was comparable with the findings of Koh et al.

However, the AULCSF in the affected eyes was 1.32 ± 0.16 in our group and exhibited a significantly lower deterioration compared to that in patients with dry eye in the above-mentioned study.

Contrast sensitivity in high spatial frequencies was affected in epiphora caused by lacrimal passage obstruction. Contrast sensitivity in low spatial frequencies are critical for orientation-mobility performance [37] such as walking or stair ascent/descent [38]. At the same time, contrast sensitivity in high spatial frequencies is also thought to have an important role for fine scale visual information in word perception [39]. So it is reasonable for patients with lacrimal passage obstruction to have difficulties in recognizing traffic signs when driving or reading books with small letters. Contrast sensitivity function in high spatial frequencies tends to worsen with the deterioration of optical quality caused by diseases, such as cataract [40], dry eye [41], and after refractive surgery [24,42]. High spatial frequency components of the retinal image significantly deteriorate with optical defocus, while low spatial frequency inputs remain relatively unchanged [43].

AULCSF was not correlated with tear meniscus parameters in this study. Similarly, Koh et al. reported that TM parameters were not correlated to quality of vision or optical quality in patients with epiphora due to nasolacrimal passage obstruction [3]. We hypothesized that quality of vision, such as contrast sensitivity, is reduced by not only TM over volume but also abnormality of tear film other than volume, such as viscosity or clarity. Hiraoka et al. reported that increased light scattering after instillation of brinzolamide cause deterioration of contrast sensitivity [11]. In patients with lacrimal passage obstruction, it is possible that light scattering may increase because of excessive retention of proteins in tear film not excreted to nasal fossa. Although there was no difference in letter contrast sensitivity between affected and contralateral eyes, the difference in letter contrast sensitivity of the affected and contralateral eyes was negatively correlated to the difference in TMH. We need to conduct a further study with a larger sample size to discuss this result.

In this study, there was a significant difference in AULCF between affected and contralateral eye. Furthermore, there were no significant differences seen in either letter contrast sensitivity or in low contrast VA. AULCSF assesses the broad contrast sensitivity function from low to high spatial frequency, while letter contrast sensitivity and low-contrast VA assess the contrast sensitivity function from limited spatial frequency. As a result, AULSCF might be more sensitive than letter contrast sensitivity and low-contrast VA [44]. Letter contrast sensitivity and low-contrast VA are simplified methods in clinical use but may not be an appropriate method to evaluate a contrast sensitivity function in patients with lacrimal passage obstruction.

One limitation of this study is that the contralateral eyes were different from the normal eyes. Although we confirmed that the lacrimal passage was intact using the lacrimal passage irrigation test, there might have been subclinical stenosis of lacrimal passage in the contralateral eye. Hence, it was meaningful to compare the parameters of the affected and contralateral eye with paired sample statistical analysis. Various types of lacrimal passage obstruction were included in this study. Future studies with larger sample sizes may analyze contrast sensitivity with each type of obstruction. Also, other ocular surface-related parameters such as blink rate should be assessed in a further study since it could impact visual function measures, including contrast sensitivity function.

In conclusion, contrast sensitivity function underwent a significant decline in eyes with epiphora caused by lacrimal passage obstruction. Thus, contrast sensitivity measurement might aid our understanding of visual disturbances in patients with lacrimal passage obstruction and also serve as one of the decisive parameters for surgical interventions in addition to a risk of dacryocystitis or developing intractable conjunctivitis.

## Acknowledgments

We would like to thank Editage (www.editage.jp) for English language editing.

## Author Contributions

**Data curation:** Kuniharu Tasaki, Sujin Hoshi, Takahiro Hiraoka.

**Supervision:** Takahiro Hiraoka, Tetsuro Oshika.

**Validation:** Takahiro Hiraoka.

**Writing – original draft:** Kuniharu Tasaki.

**Writing – review & editing:** Sujin Hoshi.

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
