## [Decision Letter · Decision Letter 0]

27 Jan 2020

PONE-D-19-33710

Deterioration of contrast sensitivity in eyes with epiphora due to lacrimal passage obstruction

PLOS ONE

Dear Dr. Hoshi,

Thank you for submitting your manuscript to PLOS ONE. After careful consideration, we feel that it has merit but does not fully meet PLOS ONE’s publication criteria as it currently stands. Therefore, we invite you to submit a revised version of the manuscript that addresses the points raised during the review process.

ACADEMIC EDITOR:

1. Please ensure all the comments raised by the reviewers are addressed.

2. A major issue in the manuscript relates to the lack of rationale - or at least evidence of the rationale - for undertaking this study. This needs to be addressed within the manuscript Introduction and Abstract. This includes for example, references to support the contention that nasolacrimal obstruction-induced epiphora impacts on visual function (including visual acuity and contrast sensitivity). Do other types of epiphora produce changes in visual function?

3. Please indicate if any medications (topical or systemic) were used by participants in the study or if this was an exclusion criteria (not mentioned in the Methods). Were participants assessed for any other ocular surface-related parameters apart from lower tear meniscus height and area? Blink rate for example, could impact visual function measures. Can the authors provide details on how long the participants had been diagnosed with epiphora and how long the nasolacrimal obstruction had been present?

4. The statistical analyses used requires further details including whether normality of results were tested for/ taken into account. Also the choice of tests used is important (for example, Figure 1 - why use a t-test when an ANOVA is more appropriate?).

5. Please provide information for pupil size during contrast sensitivity tests; the authors note pupils were not dilated but nothing on pupil sizes of affected and contralateral eyes. 

6. The Results section does not provide a complete overview of the tests undertaken, as described in the Methods. The contrast sensitivity outcomes for letters and Landolt C are mentioned in Table 1, but not described in any detail (nor in the Discussion). The same goes for the very brief mention of lower lid tear meniscus height and area. These were noted as significantly different  between affected and contralateral eyes in Table 1, but briefly noted to be not related to only AULCSF (no mention of letters or Landolt C contrast sensitivity).

7.  The underlying mechanism for the reduced contrast sensitivity is not clearly explored in the Discussion, and this would enhance the impact of the findings. 

8. The authors state in the conclusion (p. 14) that contrast sensitivity reduction may serve as a 'decisive parameter for surgical interventions'. Can the authors please include here (briefly) other critical (acute) factors taken into account in these cases where surgery is advised?  

We would appreciate receiving your revised manuscript by Mar 12 2020 11:59PM. To enhance the reproducibility of your results, we recommend that if applicable you deposit your laboratory protocols in protocols.io, where a protocol can be assigned its own identifier (DOI) such that it can be cited independently in the future. For instructions see: http://journals.plos.org/plosone/s/submission-guidelines#loc-laboratory-protocols

We look forward to receiving your revised manuscript.

Kind regards,

Michele Madigan

Academic Editor

PLOS ONE

Journal Requirements:

Reviewers' comments:

Reviewer's Responses to Questions

**Comments to the Author**

1. Is the manuscript technically sound, and do the data support the conclusions?

Reviewer #1: Yes

Reviewer #2: Yes

2. Has the statistical analysis been performed appropriately and rigorously? 

Reviewer #1: No

Reviewer #2: Yes

3. Have the authors made all data underlying the findings in their manuscript fully available?

Reviewer #1: Yes

Reviewer #2: Yes

4. Is the manuscript presented in an intelligible fashion and written in standard English?

Reviewer #1: Yes

Reviewer #2: Yes

5. Review Comments to the Author

Reviewer #1: This is interesting study focused on the influence of epiphora due to lacrimal passage obstruction on basic visual functions.

The manuscript (all the parts: introduction, method, results and disccusion) is rather short and focused on the topic indicated in the title what makes it clear. However some elements should be changed, expanded:

Introduction:

line 51: "We hypothesized that epiphora caused by lacrimal passage obstruction might affect contrast sensitivity" but why? what is the reason? what mechamism could be responsible for that?

Materials and methods:

line 64: indicate that VA of 20/20 or better on both eyes

line 71: word ALSO shoul be added before EXCLUDED

line 122-127: why Spearman's rank correlation was used but not Pearson correlation? No information about normality of data distribution is included. When t-test was used this suggests that data had normal distribution, so why Spearman test was used? Spearman test is unparametric test, so maybe some data had un-normal distribution? If some data had un-normal distribution is should be noted and not only mean but also median values should be presented in the table 2. Please explain this point.

Results:

The description of the results is rather poor. In the lines 136-137 statistical data should be added (p-val).

Line 142-145- this part is rather strange. Heading has two lines and the text only two lines. More data should be presented (figures, statistics, even if insignificant). Moreover, correlation between TMH and TMA with 12 c/deg CS should also be performed and presented.

Part in line 146-151 should not be described as a separate result part but should be included in the earlier section where AULCSF was presented, since it describe CSF but in different way-different/additional analysis.

Additionally, why t-test was used when comparing different spatial frequencies? Anova with repeated measurements should be used together with posthoc tests, but not separate t-tests in this analyses.

Discussion:

In general, there is a lack of discussion about insignificant correlation between CS and tear meniscus, If CS reduction is related to the unstable tears film, some correlation should be expected.If not, why? It should be discussed here.

More discussion should be done on the role of high spatial freq. on the every day activities since many motor activities are based on the low (peripheral vision), but not high spatial freq. This topic should be discussed more, on the base of literature.

Authors should highlight more the important role of CS test in detection of visual disturbances. This test is more sensitive than VA and should be done in patients with various tears/corneal abnormalities to detected real visual problem from others not visual symptooms.

line 163: change QUALITY OF LIFE into the QUALITY OF VISION

lines 164-166: CS is reduced in many ocular abnormalities but also after LASIK intervention (example reference: Influence of Pupil Diameter on the Relation between Ocular Higher-Order Aberration and Contrast Sensitivity after Laser In Situ Keratomileusis, Oshika et al. Investigative Ophthalmology & Visual Science, April 2006, Vol. 47, No. 4) in myopia (The Effect of Myopia on Contrast Thresholds. Bistra D. Stoimenova, Investigative Ophthalmology & Visual Science, May 2007, Vol. 48, No. 5) or with multifocal contact lenses (Katarzyna Przekoracka, et al., Contact Lens and Anterior Eye, https://doi.org/10.1016/j.clae.2019.12.002; Visual performance with simultaneous vision multifocal contact lenses, Almudena Llorente-Guillemot et. al., Clin Exp Optom 2012; 95: 1: 54–59).

Reviewer #2: Reviewer’s comments:

The study investigated the effect of epiphora on contrast sensitivity. This is an important study with case series that showed the association decrease in contrast sensitivity with epiphora. The authors have described this study clearly in their paper and it was thus easy to follow.

There are a couple of comments which will help to enhance the impact of the paper:

Comment 1: Line 91- 92: Please correct the typo error of repeated word: ‘were obtained’.

Comment 2: The introduction is very succinct and easy to follow. However, with regards to epiphora leading to decrease in QoV and quality of life, it would be good if the authors could provide the data (actual numbers) from the literature that has been referenced so that the readers can gauge the ballpark of quality of life and vision that epiphora will affect. E.g: line 45: if Vision related quality of life was significantly impaired before silicon tube intubation, then what was the actual number before and after the procedure.

Comment 3: When the affected and contralateral eyes were compared, there was a significant difference in AULCF while there were no significant differences seen in either low contrast visual acuity or in letter contrast sensitivity. It will be good to discuss the differences between these three methods in the discussion. It will also enhance the paper by highlighting how this can be done in a clinical setting.

6. PLOS authors have the option to publish the peer review history of their article (what does this mean?). If published, this will include your full peer review and any attached files.

Reviewer #1: No

Reviewer #2: Yes: Moneisha Gokhale

---

## [Author Response · Author response to Decision Letter 0]

6 Apr 2020

POINT BY POINT RESPONSE TO THE REVIEWERS’ COMMENTS

ACADEMIC EDITOR

Comment #1. Please ensure all the comments raised by the reviewers are addressed.

>Response: We made point by point response to the comments raised by the reviewers as suggested.

Comment #2. A major issue in the manuscript relates to the lack of rationale - or at least evidence of the rationale - for undertaking this study. This needs to be addressed within the manuscript Introduction and Abstract. This includes for example, references to support the contention that nasolacrimal obstruction-induced epiphora impacts on visual function (including visual acuity and contrast sensitivity). Do other types of epiphora produce changes in visual function?

>Response: Thank you for your comments. We have added the following passages and references to the manuscript to provide a more thorough rationale: 

Contrast sensitivity is reduced in epiphora induced by conjunctivochalasis (Qiu, W. et al. Evaluation of the Effects of Conjunctivochalasis Excision on Tear Stability and Contrast Sensitivity. Sci. Rep. 2016. 6, 37570) or instillation of a gel-forming solution or particle suspension (Hiraoka T, et al. Contrast Sensitivity and Optical Quality of the Eye after Instillation of Timolol Maleate Gel-Forming Solution and Brinzolamide Ophthalmic Suspension. Ophthalmology 2010. 117:11:2080-2087). As a result, we hypothesize that contrast sensitivity will also be reduced in eyes with epiphora due to lacrimal passage obstruction (lines 47-53).

abstract (line 21):

“Epiphora causes deterioration in contrast sensitivity in some eye diseases. This study was conducted to investigate contrast sensitivity in eyes with epiphora caused by lacrimal passage obstruction.”

Comment #3. Please indicate if any medications (topical or systemic) were used by participants in the study or if this was an exclusion criteria (not mentioned in the Methods). Were participants assessed for any other ocular surface-related parameters apart from lower tear meniscus height and area? Blink rate for example, could impact visual function measures. Can the authors provide details on how long the participants had been diagnosed with epiphora and how long the nasolacrimal obstruction had been present?

>Response: Patients receiving ophthalmic solution stopped using ophthalmic solution 4 weeks before examinations. We added the description to the methods (lines 68-69). 

General ocular surface parameters such as Schirmer's test or tear film breakup time are not necessarily suitable because measurement results vary in eyes with epiphora due to lacrimal passage obstruction. Since many studies investigate epiphora in NLDO measures TMH and TMA using OCT (Koh S, et al. The effect of ocular surface regularity on contrast sensitivity and straylight in dry eye. Invest Ophthalmol Vis Sci. 2017;58:2647–265. Qiu W, et al. Evaluation of the Effects of Conjunctivochalasis Excision on Tear Stability and Contrast Sensitivity. Sci. Rep 2016. 6, 37570), we also measured TMH and TMA. As suggested, blink rate could impact visual function measures, but we did not accessed blink rate in this study. We added the possible impact of blink rate to visual function measures in discussion section (line 249).

“Also, other ocular surface-related parameters such as blink rate should be assessed in a further study since it could impact visual function measures, including contrast sensitivity function.”

As suggested, We added the description about duration of epiphora in methods (line 81). “The patients presented with epiphora for a median duration of 9 (range 2-82, interquartile range 3-20.5) months.”

Comment#4. The statistical analyses used requires further details including whether normality of results were tested for/ taken into account. Also the choice of tests used is important an (for example, Figure 1 - why use a t-test when an ANOVA is more appropriate?).

>Response: As suggested, we tested the normality of the result, and some data did not have normal distribution. So we performed Wilcoxon signed-rank test in these date and corrected method (line 130) and result (line 146, 151, 152, table 2). As the reviewer suggested, ANOVA should be appropriate to analyze the difference among group means in a sample. On the other hand, Wilcoxon test for AULCSF, as kind of a group means of several log contrast sensitivity with different cpd, is generally used to compare contrast sensitivity between two groups (Hiraoka T, et al. Contrast Sensitivity and Optical Quality of the Eye after Instillation of Timolol Maleate Gel-Forming Solution and Brinzolamide Ophthalmic Suspension. Ophthalmology 2010. 117:11:2080-2087. Koh S, et al. The effect of ocular surface regu-larity on contrast sensitivity and straylight in dry eye. Invest Ophthalmol Vis Sci. 2017;58:2647–265). In this study, since we found a significant difference in AULCSF. We tried to analyze the difference of log contrast sensitivity for each cpd, so that we can have an idea which cpd is contributable.

Comment#5. Please provide information for pupil size during contrast sensitivity tests; the authors note pupils were not dilated but nothing on pupil sizes of affected and contralateral eyes. 

>Response: We did not measure pupil size, but the test was performed under photopic condition. (We added the description in method, line 101) Patients with anisocoria were excluded in this study (we added the description in method, line 75). 

“The pupils of the eyes were undilated during these monocular tests, and these tests were performed under photopic conditions.”

Comment#6. The Results section does not provide a complete overview of the tests undertaken, as described in the Methods. The contrast sensitivity outcomes for letters and Landolt C are mentioned in Table 1, but not described in any detail (nor in the Discussion). The same goes for the very brief mention of lower lid tear meniscus height and area. These were noted as significantly different between affected and contralateral eyes in Table 1, but briefly noted to be not related to only AULCSF (no mention of letters or Landolt C contrast sensitivity).

>Response: We added the p-value of the contrast sensitivity outcomes for letters and Landolt C in the results (line 151, 152) and discussed the differences between three contrast sensitivity measurements in the discussion (line 232). 

“In this study, there was a significant difference in AULCF between affected and contralateral eye. Furthermore, there were no significant differences seen in either letter contrast sensitivity or in low contrast VA. AULCSF assesses the broad contrast sensitivity function from low to high spatial frequency, while letter contrast sensitivity and low-contrast VA assess the contrast sensitivity function from limited spatial frequency. As a result, AULSCF might be more sensitive than letter contrast sensitivity and low-contrast VA (Parede TRR, et al. Quality of vision in refractive and cataract surgery, indirect measurers: review article. Arq Bras Oftalmol. 2013;76(6):386-90). Letter contrast sensitivity and low-contrast VA are simplified methods in clinical use, but may not be an appropriate method to evaluate a contrast sensitivity function in patients with lacrimal passage obstruction.”

We added the description about correlation between tear meniscus parameters and letters or Landolt C contrast sensitivity in the result (line 170). 

“Although there was no difference in letter contrast sensitivity between affected and contralateral eyes, the difference in letter contrast sensitivity of the affected and contralateral eyes was negatively correlated to the difference in TMH. We need to conduct a further study with larger sample size to discuss this result.” (line 228)

Comment#7. The underlying mechanism for the reduced contrast sensitivity is not clearly explored in the Discussion, and this would enhance the impact of the findings. 

>Response:

AULCSF was not correlated with TM parameters. So we hypothesized that contrast sensitivity is reduced by not only TM over volume but also abnormality of tear film other than volume, such as viscosity or clarity. Hiraoka et al. report that increased light scattering after instillation of brinzolamide cause deterioration of contrast sensitivity (Hiraoka T, et al. Contrast Sensitivity and Optical Quality of the Eye after Instillation of Timolol Maleate Gel-Forming Solution and Brinzolamide Ophthalmic Suspension. Ophthalmology 2010. 117:11:2080-2087). In patients with lacrimal passage obstruction, it is possible that light scattering may increase because of excessive retention of proteins in tear film not excreted to nasal fossa. We added this in the discussion section (line 222).

Comment#8. The authors state in the conclusion (p. 14) that contrast sensitivity reduction may serve as a 'decisive parameter for surgical interventions'. Can the authors please include here (briefly) other critical (acute) factors taken into account in these cases where surgery is advised? 

>Response: Surgery is also considered when patients are at risk of dacryocystitis or developing intractable conjunctivitis. We added the description to the discussion (line 256).

Reviewer #1

Comment #1: This is interesting study focused on the influence of epiphora due to lacrimal passage obstruction on basic visual functions. The manuscript (all the parts: introduction, method, results and disccusion) is rather short and focused on the topic indicated in the title what makes it clear. However some elements should be changed, expanded:

>Response: 

Thank you for your comments. Folloing your instructions, I have changed and expanded some elements in the manuscript.

Comment#2:

Introduction:

line 51: "We hypothesized that epiphora caused by lacrimal passage obstruction might affect contrast sensitivity" but why? what is the reason? what mechamism could be responsible for that?

>Response:

We have added expanded on the rationale and hypothesis of the study. 

Contrast sensitivity is reduced in epiphora induced by conjunctivochalasis (Qiu, W. et al. Evaluation of the Effects of Conjunctivochalasis Excision on Tear Stability and Contrast Sensitivity. Sci. Rep. 2016. 6, 37570) or instillation of a gel-forming solution or particle suspension (Hiraoka T, et al. Contrast Sensitivity and Optical Quality of the Eye after Instillation of Timolol Maleate Gel-Forming Solution and Brinzolamide Ophthalmic Suspension. Ophthalmology 2010. 117:11:2080-2087). As a result, we hypothesize that contrast sensitivity will also be reduced in eyes with epiphora due to lacrimal passage obstruction (lines 47-53).

abstract (line 21):

“Epiphora causes deterioration in contrast sensitivity in some eye diseases. This study was conducted to investigate contrast sensitivity in eyes with epiphora caused by lacrimal passage obstruction.”

Materials and methods:

line 64: indicate that VA of 20/20 or better on both eyes

>Response: Thank you for your review. We added “in both eyes” in after “VA of 20/20 or better”.

line 71: word ALSO should be added before EXCLUDED

>Response: We added “also” before “excluded”.

line 122-127: why Spearman's rank correlation was used but not Pearson correlation? No information about normality of data distribution is included. When t-test was used this suggests that data had normal distribution, so why Spearman test was used? Spearman test is unparametric test, so maybe some data had un-normal distribution? If some data had un-normal distribution is should be noted and not only mean but also median values should be presented in

the table 2. Please explain this point.

>Response: Thank you for your careful review. It was our mistake to describe Spearman’s rank correlation. We actually used Pearson correlation coefficient, and revised the method (line 135). Some of our data had un-normal distribution, so we also revised methods (line 130) and the result (line 146, 148, 151, table2) to clarify this.

Results:

The description of the results is rather poor. In the lines 136-137 statistical data should be added (p-val). Line 142-145- this part is rather strange. Heading has two lines and the text only two lines. More data should be presented (figures, statistics, even if insignificant). Moreover, correlation between TMH and TMA with 12 c/deg CS should also be performed and presented.

Part in line 146-151 should not be described as a separate result part but should be included in the earlier section where AULCSF was presented, since it describe CSF but in different way different/additional analysis. Additionally, why t-test was used when comparing different spatial frequencies? Anova with repeated measurements should be used together with posthoc tests, but not separate t-tests in this analyses.

>Response:

Following your instructions, we added p-values in results. 

As suggested, we have added more detail in the last part of result section.

We changed the title of this part as follows: Correlations between the difference in tear meniscus and the difference in quality of vision of affected and contralateral eyes

As suggested, we checked the correlation between TMH and TMA with 12 cpd CS, and found no correlation (line169).

Also, following the editor's suggestion, we added the results of the correlation between TMH and TMA with other contrast sensitivity measurements and BCVA (line 170).

As suggested, a paragraph describing contrast sensitivity at four specific frequencies for the affected and contralateral eyes has been moved in the earlier section where AULCSF was presented (line 147).

As suggested, ANOVA should be appropriate to analyze the difference among group means in a sample. On the other hand, Wilcoxon test for AULCSF, as kind of a group means of several log contrast sensitivity with different cpd, is generally used to compare contrast sensitivity between two groups [Hiraoka T, et al. Contrast Sensitivity and Optical Quality of the Eye after Instillation of Timolol Maleate Gel-Forming Solution and Brinzolamide Ophthalmic Suspension. Ophthalmology 2010. 117:11:2080-2087. Koh]. In this study, since we found a significant difference in AULCSF. We also attempted to analyze the difference of log contrast sensitivity for each cpd, to gain insight into which cpd contributes to this difference.

Discussion:

In general, there is a lack of discussion about insignificant correlation between CS and tear meniscus. If CS reduction is related to the unstable tears film, some correlation should be expected. If not, why? It should be discussed here.

>Response:

As the reviewer indicated, we added a description to discuss the insignificant correlation between CS and tear meniscus. (line 222)

“We hypothesized that quality of vision, such as contrast sensitivity, is reduced by not only TM over volume but also abnormality of tear film other than volume, such as viscosity or clarity. Hiraoka et al. reported that increased light scattering after instillation of brinzolamide cause deterioration of contrast sensitivity [35]. In patients with lacrimal passage obstruction, it is possible that light scattering may increase because of excessive retention of proteins in tear film not excreted to nasal fossa.”

More discussion should be done on the role of high spatial freq. on the every day activities since many motor activities are based on the low (peripheral vision), but not high spatial freq. This topic should be discussed more, on the base of literature. 

　Authors should highlight more the important role of CS test in detection of visual disturbances. This test is more sensitive than VA and should be done in patients with various tears/corneal abnormalities to detected real visual problem from others not visual symptoms.

>Response: Following your advice, we added the description in the discussion. Contrast sensitivity in high spatial frequencies is thought to have an important role for fine-scale visual information in word perception (Geoffrey R, et al. Spatial Frequency Sensitivity Differences between Adults of Good and Poor Reading Ability. Invest Ophthalmol Vis Sci. 2005;46:2219 –2224). As a result, it is reasonable for patients with lacrimal passage obstruction to have difficulties in recognizing traffic sign when driving or reading books with small letters. (line 213)

Contrast sensitivity in low spatial frequencies is critical for orientation-mobility performance [Marron JA, et al. Visual factors and orientation-mobility performance. Am J Optom Physiol Opt. 1982;59:413-426] such as walking or stair ascent/descent [West et al. Arch Ophthalmol 2002;120:774-780]. We added this in the discussion section (line 213).

As suggested, we added description about the importance of CS in detection of visual desturbances. (line 188)

line 163: change QUALITY OF LIFE into the QUALITY OF VISION

Authours’ response: we changed quality of life into the quality of vision (line 185)

lines 164-166: CS is reduced in many ocular abnormalities but also after LASIK intervention (example reference: Influence of Pupil Diameter on the Relation between Ocular Higher-Order Aberration and Contrast Sensitivity after Laser In Situ Keratomileusis, Oshika et al. Investigative Ophthalmology & Visual Science, April 2006, Vol. 47, No. 4) in myopia (The Effect of Myopia on Contrast Thresholds. Bistra D. Stoimenova, Investigative Ophthalmology & Visual Science, May 2007, Vol. 48, No. 5) or with multifocal contact lenses (Katarzyna Przekoracka, et al., Contact Lens and Anterior Eye, https://doi.org/10.1016/j.clae.2019.12.002; Visual performance with simultaneous vision multifocal contact lenses, Almudena Llorente-Guillemot et. al., Clin Exp Optom 2012; 95: 1: 54–59).

>Response: Following your advice, we added the sentences in the discussion section (line 187) and references.

Reviewer #2: Reviewer’s comments:

The study investigated the effect of epiphora on contrast sensitivity. This is an important study with case series that showed the association decrease in contrast sensitivity with epiphora. The authors have described this study clearly in their paper and it was thus easy to follow.

There are a couple of comments which will help to enhance the impact of the paper:

Comment 1: Line 91- 92: Please correct the typo error of repeated word: ‘were obtained’. 

>Response:

Thank you for your comments. Following your instructions, I have corrected the typo error of repeated word: ‘were obtained’. (line 100)

Comment 2: The introduction is very succinct and easy to follow. However, with regards to epiphora leading to decrease in QoV and quality of life, it would be good if the authors could provide the data (actual numbers) from the literature that has been referenced so that the readers can gauge the ballpark of quality of life and vision that epiphora will affect. 

E.g., line 45: If vision-related quality of life was significantly impaired before silicon tube intubation, then what was the actual number before and after the procedure.

>Response:

Composite score from the 25-item national eye institute visual function questionnaire（NEIVFQ-25）improved 76.3 ± 11.5 to 82.0 ± 11.3, p=0.001. We added the description in the introduction section. (line 48)

Comment 3: When the affected and contralateral eyes were compared, there was a significant difference in AULCF while there were no significant differences seen in either low contrast visual acuity or in letter contrast sensitivity. It will be good to discuss the differences between these three methods in the discussion. It will also enhance the paper by highlighting how this can be done in a clinical setting.

>Response:

We added this content to the discussion section (line 237).

AULCSF assesses the broad contrast sensitivity function from low to high spatial frequency, while letter contrast sensitivity and low-contrast VA assess the contrast sensitivity function from limited spatial frequency. As a result, AULSCF might be more sensitive than letter contrast sensitivity and low-contrast VA (Parede TRR, Torricelli AMM, Mukai A, Netto MV, Bechara SJ, Quality of vision in refractive and cataract surgery, indirect measurers: review article. Arq Bras Oftalmol. 2013;76(6):386-90). Letter contrast sensitivity and low-contrast VA are simplified methods in clinical use but may not be an appropriate method to evaluate a contrast sensitivity function in patients with lacrimal passage obstruction.

---

## [Decision Letter · Decision Letter 1]

4 May 2020

Deterioration of contrast sensitivity in eyes with epiphora due to lacrimal passage obstruction

PONE-D-19-33710R1

Dear Dr. Hoshi,

We are pleased to inform you that your manuscript has been judged scientifically suitable for publication and will be formally accepted for publication once it complies with all outstanding technical requirements.

With kind regards,

Michele Madigan

Academic Editor

PLOS ONE

Additional Editor Comments (optional):

Reviewers' comments:

Reviewer's Responses to Questions

**Comments to the Author**

1. If the authors have adequately addressed your comments raised in a previous round of review and you feel that this manuscript is now acceptable for publication, you may indicate that here to bypass the “Comments to the Author” section, enter your conflict of interest statement in the “Confidential to Editor” section, and submit your "Accept" recommendation.

Reviewer #1: (No Response)

Reviewer #2: All comments have been addressed

2. Is the manuscript technically sound, and do the data support the conclusions?

Reviewer #1: Yes

Reviewer #2: Yes

3. Has the statistical analysis been performed appropriately and rigorously? 

Reviewer #1: Yes

Reviewer #2: Yes

4. Have the authors made all data underlying the findings in their manuscript fully available?

Reviewer #1: Yes

Reviewer #2: Yes

5. Is the manuscript presented in an intelligible fashion and written in standard English?

Reviewer #1: Yes

Reviewer #2: Yes

6. Review Comments to the Author

Reviewer #1: The authors have adequately addressed your comments raised in a previous round but some detail correction should be made:

1) spaces: sometimes the are space between +, -, + but sometimes not - please correct it

2) there is no consistency in writing numbers with p-value, sometimes 2 numerbs after dot is given, but sometimes 3 numebr - I reccomend to use always 3 numbers

3) references has been added but one of them was omitted: Cont Lens Anterior Eye. 2020 Feb;43(1):33-39. doi: 10.1016/j.clae.2019.12.002. Epub 2019 Dec 13. "Contrast sensitivity and visual acuity in subjects wearing multifocal contact lenses with high additions designed for myopia progression control." I recommend to add this reference since it is very up to date (2020) and show interesting data on the central and peripgeral CS.

Reviewer #2: The manuscript is now sound. Reviewer is happy with the changes made by the authors in response to the comments and feedback.

No more comments.

7. PLOS authors have the option to publish the peer review history of their article (what does this mean?). If published, this will include your full peer review and any attached files.

Reviewer #1: No

Reviewer #2: Yes: Moneisha Gokhale

---

## [Editor Report · Acceptance letter]

8 May 2020

PONE-D-19-33710R1 

Deterioration of contrast sensitivity in eyes with epiphora due to lacrimal passage obstruction 

Dear Dr. Hoshi:

I am pleased to inform you that your manuscript has been deemed suitable for publication in PLOS ONE. Congratulations! Your manuscript is now with our production department. 

With kind regards,

on behalf of

Dr. Michele Madigan 

Academic Editor

PLOS ONE